# Divergence-enhanced Knowledge-guided Context Optimization for Visual-Language Prompt Tuning

**Yilun Li, Miaomiao Cheng✉, Xu Han, Wei Song✉**
College of Information Engineering, Capital Normal University, Beijing, China
{Yilun, miaomiao, hanxu, wsong}@cnu.edu.cn

## Abstract

Prompt tuning vision-language models like CLIP has shown great potential in learning transferable representations for various downstream tasks. The main issue is how to mitigate the over-fitting problem on downstream tasks with limited training samples. While knowledge-guided context optimization has been proposed by constructing consistency constraints to handle catastrophic forgetting in the pre-trained backbone, it also introduces a bias toward pre-training. This paper proposes a novel and simple Divergence-enhanced Knowledge-guided Prompt Tuning (DeKg) method to address this issue. The key insight is that the bias toward pre-training can be alleviated by encouraging the independence between the learnable and the crafted prompt. Specifically, DeKg employs the Hilbert-Schmidt Independence Criterion (HSIC) to regularize the learnable prompts, thereby reducing their dependence on prior general knowledge, and enabling divergence induced by target knowledge. Comprehensive evaluations demonstrate that DeKg serves as a plug-and-play module that can seamlessly integrate with existing knowledge-guided context optimization methods and achieves superior performance in three challenging benchmarks. We make our code available at https://github.com/cnunlp/DeKg.

## 1 Introduction

Large-scale vision-language models (VLMs) like CLIP (Radford et al., 2021) and ALIGN (Jia et al., 2021) have demonstrated excellent capabilities in zero-shot recognition and generalization representation. Unfortunately, the large model sizes, high computational resource requirements, and massive trainable data restrict their deployment on real vision-language tasks. To address this problem, a new paradigm of prompt tuning has been proposed and attracted increasing attention in recent years (Radford et al., 2021; Zhou et al., 2022b).

Prompt tuning (Zhou et al., 2022b) aims to optimize a limited set of dynamic continuous prompt representations with the end-to-end objective function, i.e., the cross-entropy loss, to transfer the pre-trained knowledge of VLMs to targeted tasks. These methods are less than optimal due to challenges in determining what should be preserved and what should be adapted for downstream tasks. For example, in the base-to-new generalization task, as shown in Figure 1, CoOp (Zhou et al., 2022b) can achieve a significant performance improvement over the manually prompted method CLIP (Radford et al., 2021) on base (seen) classes, yet is inferior on new (unseen) classes in the same dataset. This suggests that the prior general knowledge may be distorted by the limited task-specific labeled data, causing fine-tuned models to deviate from the pre-trained VLMs and leading to overfitting issues.

The overfitting issues can be attributed to the lack of regularization in the latent space to model the prior general knowledge for the unseen class distribution (Yao et al., 2023). Since the frozen CLIP (Radford et al., 2021) coupled with crafted prompts exhibits robust abilities to unseen classes, indicating that the pre-trained backbone serves as a valuable source of prior knowledge for each class, recent works (Yao et al., 2023; Zhu et al., 2023a;b; Yao et al., 2024) all construct a novel

---

✉ Corresponding authors.

constraint term by enforcing the consistency between the learnable and crafted prompts, called knowledge-guided context optimization (KGCO). However, despite the benefits of regularization in preventing catastrophic forgetting, KGCO tends to be biased toward the pre-trained model, especially when the data distribution of the target task differs from that of the pre-trained data. For example, as shown in Figure 1, KgCoOp (Yao et al., 2023) improves CoOp on new classes but degrades on base classes, mainly due to the bias of the learnable prompts toward the representations of the pre-trained CLIP. Overall, an effective prompt tuning method should address the contradiction problem between catastrophic forgetting in fine-tuning and bias in pre-training.

In this work, we propose a novel method, called Divergence-enhanced Knowledge-guided Prompt tuning (**DeKg**). We aim to maintain the advantage of knowledge-guided context optimization but alleviate the contradiction problem between catastrophic forgetting and bias towards general knowledge. Specifically, we introduce a novel constraint by employing the Hilbert-Schmidt Independence Criterion (HSIC) regularization (Gretton et al., 2005) to ensure independence between learnable and crafted prompts. The proposed model encourages the learnable prompts to maintain a consistent yet independent relation with general knowledge, optimizing the balance between adapting general knowledge and fine-tuning for targeted tasks. As shown in Figure 1, DeKg overcomes the weakness of KgCoOp, performing best on both base classes and new classes.

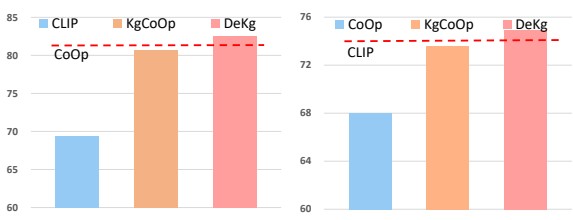

(a) Base class performance.  (b) New class performance.

Figure 1: Performance comparison of DeKg with prompt tuning methods (CLIP/ CoOp, KgCoOp (baseline method), and DeKg (Ours)) under base-to-new generalization setting. We measure average accuracy on the base classes (a) and new classes (b) over 11 datasets. The red dotted line indicates the performance of CoOp for base classes and the zero-shot CLIP for new classes.

Our contributions can be summarized as follows:

- We tackle an inherent issue of knowledge-guided context optimization in overly biasing general knowledge in pre-training, and propose a novel HSIC-based regularization method DeKg for encouraging independence between the learnable and the crafted prompts.

- DeKg integrates seamlessly with existing knowledge-guided methods. Compared to the baselines, DeKg not only introduces divergence between the learnable and crafted prompts but also enhances differentiation between learnable prompts for distinct classes.

- Extensive experiments demonstrate the superiority of the proposed method in three challenging benchmarks: base-to-new generalization, cross-dataset generalization, and few-shot learning.

## 2  RELATED WORK

Vision-Language Models (VLMs) pre-trained on large-scale image-text association pairs through self-supervised methods have exhibited impressive performance in various visual tasks (Radford et al., 2021; Jia et al., 2021). Despite the powerful generalization capacities, the enormous size of these models makes it challenging to fine-tune the entire models for downstream tasks, particularly when dealing with few-shot data. Such a trend raises the essential need to study different adaptation approaches, where prompting has been shown to be one of the simple and effective strategies.

**Prompt Tuning for VLMs**: Prompting was initially proposed in the domain of Natural Language Processing (NLP) (Lester et al., 2021; Li & Liang, 2021), providing textual instructions to the task input for distilling task-relevant knowledge. For example, CLIP (Radford et al., 2021) utilizes a collection of crafted templates "a photo of a [CLASS]" as textual inputs for category-wise embeddings, and demonstrates exceptional zero-shot image recognition capabilities. However, building a proper predefined prompt requires domain-specific knowledge and enormous time. To circumvent this, a series of methods that automate learning embeddings at the input tokens, known as soft prompts,

have emerged for fast adaptation to various downstream tasks. CoOp (Zhou et al., 2022b) optimizes the prompt content by a continuous set of learnable vectors that are used as input to the text encoder alongside the class name. However, the prompts are learned by minimizing the classification error on a training set within the given base classes, resulting in weak generalization on new classes. Co-CoOp (Zhou et al., 2022a) further expands by constructing conditional prompts on specific image instances. However, such methods have a worse generalization than CLIP on the same task to the unseen classes. In addition to the textual prompt tuning, MaPLe (Khattak et al., 2023a) conducts the visual-textual prompt tuning by jointly conducting the prompt tuning on the visual and text encoders.

**Knowledge-guided Prompt Tuning**: To ensure that learnable prompts retain essential general textual knowledge contained in frozen CLIP, ProGrad (Zhu et al., 2023a) and KgCoOp (Yao et al., 2023) both constrain the consistency between the learnable prompt and the crafted prompt by employing a novel constraint term. Specifically, ProGrad tries to optimize the learnable prompts with the aligned direction generated by the crafted prompts. KgCoOp adopts the Euclidean distance to minimize the discrepancy between textual embeddings generated by learned prompts and crafted prompts. PromptSRC (Khattak et al., 2023b) presents a self-regulating approach to prompt learning, overcoming overfitting and improving generalization by leveraging mutual agreement, prompt self-ensembling, and textual diversity. Later, TCP (Yao et al., 2024) constructs an embedding module to inject the class-level textual knowledge into the learnable prompt tokens. While existing prompt learning techniques have boosted the generalization ability by applying consistency constraints on the textual input between learnable and crafted tokens, they exhibit limited capability to capture specific knowledge. To mitigate this limitation, we propose a novel textual prompting method that incorporates consistency and diversity to enhance the generalization and discriminative capabilities of the learnable tokens.

## 3 METHODOLOGY

Our method is built upon the framework of knowledge-guided context optimization (Yao et al., 2023), which enforces a consistency constraint between the learnable and crafted prompts to distill knowledge from the frozen encoders, thus defying catastrophic forgetting. However, relying too much on pre-trained knowledge may hurt downstream knowledge and degrade performance. To mitigate this limitation, we propose a new method based on the Hilbert-Schmidt Independence Criterion (HSIC) regularization, to empower the capabilities of capturing task-specific information without forgetting task-agnostic general knowledge.

### 3.1 REVISITING KNOWLEDGE-GUIDED CONTEXT OPTIMIZATION

CLIP (Radford et al., 2021) is a fundamental Vision-Language Model, offering a zero-shot transfer strategy by pre-training the visual backbone and textual encoder on 400M large-scale image-text pairs through contrastive learning. Benefiting its robust generalization capabilities, the frozen textual embeddings $\mathbf{W}^{clip} = \{\mathbf{w}_i^{clip}\}_{i=1}^{N_c}$ of the crafted prompt "a photo of a [class]" can be a valuable source of prior general knowledge[1], where "[class]" is replaced by one of the $N_c$ class names. However, general knowledge is less able to accurately describe downstream tasks, mainly without considering the task-specific knowledge of each task.

To obtain discriminative target task knowledge, a sequence of learnable tokens $\mathbf{t} = \{\mathbf{t}_1, \mathbf{t}_2, \ldots, \mathbf{t}_M\}$ is designed for generating task-specific textual embeddings of all classes, where $M$ is the number of tokens. The corresponding class token $\mathbf{c}_i$ is concatenated with the learnable tokens, i.e., $\{\mathbf{t}_1, \mathbf{t}_2, \ldots, \mathbf{t}_M, \mathbf{c}_i\}$, for generating the encoded textual embedding $\mathbf{w}_i$. Then the learnable prompts (or contexts) $\mathbf{W} = \{\mathbf{w}_i\}_{i=1}^{N_c}$ of all classes can be optimized by minimizing the contrastive loss between the given image's embedding $\mathbf{x}$ and its class embedding $\mathbf{w}_y$, which can be formulated as

$$\mathcal{L}_{ce} = \frac{1}{N} \sum_{(\mathbf{x},y) \in \mathcal{D}_s} \frac{\exp(\mathbf{sim}(\mathbf{x}, \mathbf{w}_y)/\tau)}{\sum_{i=1}^{N_c} \exp(\mathbf{sim}(\mathbf{x}, \mathbf{w}_i)/\tau)}, \tag{1}$$

where $\mathcal{D}_s$ denotes the seen dataset, $N$ is the number of training images, $N_c$ is the number of classes, $\mathbf{sim}(\cdot)$ represents the cosine similarity, and $\tau$ refers to a temperature parameter frozen in CLIP.

---

[1]Following Yao et al. (2023), "general knowledge" in this work denotes the information contained in the pre-trained CLIP model.

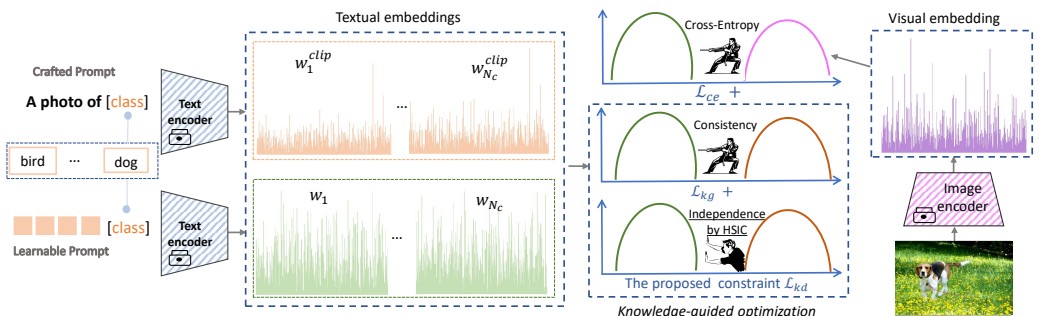

Figure 2: The knowledge-guided context optimization framework of DeKg. $\mathcal{L}_{ce}$ is the cross-entropy loss, and $\mathcal{L}_{kg}$ is a consistency constraint. $\mathcal{L}_{kd}$ is a regularization term that uses the Hilbert-Schmidt Independence Criterion (HSIC) to encourage the independence between learnable and crafted prompts.

Despite delivering promising results, it can be observed that the learned prompt is prone to overfitting to small training data and weakens the generalization capabilities to new classes (Zhou et al., 2022a), mainly because the prompt is fixed once learned and only optimized for specific classes, i.e., catastrophic forgetting for pre-trained knowledge. To exploit the prior general knowledge contained in the frozen CLIP for learnable tokens optimization, a simple yet efficient consistency constraint is added during prompt tuning to prevent catastrophic forgetting (Yao et al., 2023), which can be expressed as

$$\mathcal{L}_{kg} = \frac{1}{N_c} \sum_{i=1}^{N_c} \|\mathbf{w}_i - \mathbf{w}_i^{clip}\|_2^2. \tag{2}$$

The consistency constraint enforces that the learnable tokens have similar distributions as the crafted prompts, suggesting the potential bias toward pre-training. The reason lies in that data distributions vary across different domains. Compared to the pre-trained VLMs, the training data of downstream tasks is extremely limited, resulting in the learnable prompts inevitably towards pre-trained knowledge distributions.

## 3.2 Divergence-enhanced Knowledge-guided Context Optimization

Based on the consistency constraint employed in the knowledge-guided context optimization (KGCO) methods (Yao et al., 2023; 2024), the learnable tokens aim to preserve the task-agnostic general knowledge, enhancing the capabilities of new class prediction. However, the classifier generated by such textual tokens has a poor task-specific discriminative ability to describe downstream tasks. To alleviate the limitations of the KGCO methods, we propose a Divergence-enhanced Knowledge-guided context optimization (DeKg) strategy by introducing an independence regularization into the KGCO methods, to adapt the pre-trained CLIP to the downstream tasks. As shown in Figure 2, DeKg designs the independence constraint to enhance the divergence between learnable and crafted prompts, thereby strengthening task-specific knowledge to avoid bias towards pretraining and improving the discriminative ability of downstream tasks. Therefore, the independence constraint can act as a complement to the consistency constraint, enhancing the overall KGCO.

Given the textual embeddings embodied general knowledge $\mathbf{W}^{clip} = \{\mathbf{w}_i^{clip}\}_{i=1}^{N_c}$ with $N_c$ classes, the independence constraint is proposed to enable learnable embeddings $\mathbf{W} = \{\mathbf{w}_i\}_{i=1}^{N_c}$ to capture task-specific discriminative knowledge without interference from prior general knowledge. With the benefit of non-parametric, easy computability, rapid convergence, and small estimation bias with finite samples (Ma et al., 2020), the Hilbert-Schmidt Independence Criterion (HSIC) (Gretton et al., 2005) is adopted to penalize the dependency between the learnable and crafted prompts. Specifically, it measures the degree of dependency, with lower values indicating stronger independence and higher values suggesting a greater correlation. The proposed constraint can be formulated as

$$\mathcal{L}_{kd} = \text{HSIC}(\mathbf{W}, \mathbf{W}^{clip}) = (N_c - 1)^{-2} tr(\mathbf{K}\mathbf{H}\mathbf{K}^{clip}\mathbf{H}), \tag{3}$$

where $\mathbf{K} \in \mathbb{R}^{N_c \times N_c}$, $\mathbf{K}^{clip} \in \mathbb{R}^{N_c \times N_c}$ with entries $\mathbf{K}_{i,j} = k(\mathbf{w}_i, \mathbf{w}_j)$ and $\mathbf{K}_{i,j}^{clip} = k(\mathbf{w}_i^{clip}, \mathbf{w}_j^{clip})$, $k(\cdot, \cdot)$ is a kernel function; $\mathbf{H} = \mathbf{I}_{N_c} - \frac{1}{N_c} \mathbf{1}_{N_c} \mathbf{1}_{N_c}^T \in \mathbb{R}^{N_c \times N_c}$ is the centering matrix, which is used to remove the bias within each representation and focus on the inter-variable relationships; $tr(\cdot)$ represents the trace of the matrix.

Define $\mathbf{A} = \mathbf{H}\mathbf{K}^{clip}\mathbf{H}$, then Eq.(3) can be rewritten as follows

$$
\begin{aligned}
\mathcal{L}_{kd} &= (N_c - 1)^{-2} tr(\mathbf{K}\mathbf{A}) \\
&= (N_c - 1)^{-2} \sum_{i,j} \mathbf{K}_{i,j} \mathbf{A}_{i,j}.
\end{aligned}
\tag{4}
$$

Notice that the elements $\mathbf{A}_{i,j}$ indicate the inter-class relationships among the crafted prompts $\mathbf{W}^{clip}$ and are fixed, since they solely relate to the representations of the crafted prompts. Consequently, $\mathcal{L}_{kd}$ is influenced only by $\{\mathbf{K}_{i,j}\}$ which describes the relationships within the set of learnable prompts $\mathbf{W}$. Therefore, we can identify two advantages of $\mathcal{L}_{kd}$.

First, the computation is only related to $\mathbf{W}$ without introducing any extra parameters. In our implementation, we use the inner product kernel function, i.e., $\mathbf{K}_{i,j} = \mathbf{w}_i^T \mathbf{w}_j$, and promising performance is achieved.

Second, leveraging HSIC regularization, $\mathcal{L}_{kd}$ fosters the independence between the learnable and crafted prompts by enhancing the divergence among the learnable prompts across various classes, which is influenced by the inter-class relationships inherent in the crafted prompts. So $\mathcal{L}_{kd}$ helps capture task-specific unique knowledge, to enhance the model's discriminative capability.

Remind that the consistency constraint $\mathcal{L}_{kg}$ aims to maximize the agreement between learnable and crafted prompts, to enhance the model's generalization capability. Therefore, $\mathcal{L}_{kd}$ and $\mathcal{L}_{kg}$ focus on optimizing different aspects and complement each other.

As a result, we constrain the learnable prompts with both consistency and independence. The final objective function can be expressed as

$$
\mathcal{L} = \mathcal{L}_{ce} + \lambda \mathcal{L}_{kg} + \mu \mathcal{L}_{kd},
\tag{5}
$$

where $\lambda$ and $\mu$ are trade-off hyperparameters encoding the belief degrees for consistency and independence constraints, respectively.

## 4 EXPERIMENTS

In this section, we conduct extensive experiments on three widely-used benchmarks to evaluate the ability of base-to-new generalization, cross-data generalization, and few-shot learning, and demonstrate the effectiveness of the proposed method by comparing with strong vision-language prompt tuning baselines.

### 4.1 EXPERIMENTAL SETUP

**Datasets**: For downstream tasks, we follow previous work (Radford et al., 2021; Zhou et al., 2022a;b), to conduct experiments on 11 representative image classification datasets, including ImageNet (Deng et al., 2009) and Caltech (Fei-Fei et al., 2004) for generic object classification; Oxford-Pets (Parkhi et al., 2012), StanfordCars (Krause et al., 2013), Flowers (Nilsback & Zisserman, 2008), Food101 (Bossard et al., 2014), and FGVCAircraft (Maji et al., 2013) for fine-grained visual categorization, EuroSAT (Helber et al., 2019) for satellite image classification, UCF101 (Soomro et al., 2012) for action recognition, DTD (Cimpoi et al., 2014) for texture classification, and SUN397 (Xiao et al., 2010) for scene recognition.

**Baselines**: First, to demonstrate that DeKg can embody the advantage of preserving both the general knowledge frozen in CLIP and task-specific knowledge, we compare the results of CLIP (Radford et al., 2021), CoOp (Zhou et al., 2022b), CoCoOp (Zhou et al., 2022a), and MaPLe (Khattak et al., 2023a), which exploits only general or task-specific knowledge, i.e., only uses cross-entropy for prediction. Second, to show the significant advantage of enhancing task-specific knowledge, we compared with two baselines, KgCoOp (Yao et al., 2023) and ProGrad (Zhu et al., 2023a), which

preserve the general knowledge by enforcing the consistency between the learnable tokens and crafted prompts. Besides, to highlight the importance of divergence guided by general knowledge and task-specific knowledge, we compared with PromptSRC (Khattak et al., 2023b) and TCP (Yao et al., 2024), which incorporate other strategies to consistency constraint, i.e., PromptSRC adds self-ensembling and textual diversity regularization, while TCP inserts class-specific knowledge into embeddings.

For the DeKg method which unifies the general knowledge preservation and divergence upon general-specific knowledge into one framework, four baselines, i.e., KgCoOp, ProGrad, TCP, and PromptSRC, can be expanded by adding the HSIC regularization to produce the divergence by target knowledge with general knowledge preservation. In our experiments, only KgCoOp and TCP are adopted and expanded to generate the final learnable tokens, denoted as $DeKg_{KgCoOp}$ and $DeKg_{TCP}$ respectively. The main reason for this is that, on one hand, ProGrad aligns prompts with general knowledge of the gradient, while the others are directly aligned with the embeddings. On the other hand, PromptSRC includes visual prompts and textual prompts, while other baselines only include textual prompts.

**Training Details**: Our implementation is based on KgCoOp's (Yao et al., 2023) and TCP's (Yao et al., 2024) codes. To ensure a fair comparison, all experiments were conducted using the ViT-B/16 (Dosovitskiy et al., 2021) as the vision backbone and the context length set as 4. Additionally, we maintained consistency with the corresponding baselines in $DeKg_{KgCoOp}$ and $DeKg_{TCP}$ for random prompt initialization, training epoch, training schedule, and data augmentation settings. In our experiments, we set the ratio of $\lambda/\mu$ to 3/1 by grid search, which translates to $\lambda$ being 6 and $\mu$ being 2. All experiments were carried out using the HYGON DCU-Z100L server.

## 4.2 PERFORMANCE COMPARISON AND ANALYSIS

### 4.2.1 BASE-TO-NEW GENERALIZATION

The base-to-new generalization setting aims to evaluate whether the models learned on base tasks can generalize to new tasks without unseen classes, i.e., a *category shift* exists between base and new tasks. Following the baselines, on each dataset, we first construct a base and new task by equally dividing the dataset into two groups, then perform prompt tuning on the base classes and test the learned model on both the base and new tasks. Table 1 presents the performance of different methods across 11 datasets with 16-shot samples, where the best and second results are marked in bold and underlined, respectively. For convenience, we refer to the classification accuracy of base tasks and new tasks as base accuracy and new accuracy, respectively. The harmonic mean (H) of base accuracy and new accuracy is also computed to demonstrate the generalization trade-off.

Compared with zero-shot CLIP, the baselines optimized with only cross-entropy loss, i.e., CoOp, CoCoOp, and MaPLe, achieve improvement on base classes but show inferior performance on new classes except MaPLe. This suggests that they overall tend to overfit the task-specific data distributions, losing the original generalization capability of the frozen CLIP model towards new tasks. Although KgCoOp alleviates the poor generalization problem in CoOp by preserving the prior general knowledge, it hardly outperforms CoOp in base accuracy in almost all benchmarks, i.e., KgCoOp has an average drop from $82.64\%$ to $80.73\%$ compared with CoOp, while ProGrad has a similar trend. This suggests that the learnable prompts may be skewed towards the general knowledge frozen in the CLIP, due to the limited task-specific knowledge. In contrast, DeKg improves on both base and new classes over CLIP and CoOp. Specifically, $DeKg_{TCP}$ obtains an average gain of $2.32\%$ (i.e., $84.96\%$ vs. $82.64\%$) over CoOp in base accuracy, and $2.16\%$ (i.e., $74.22\%$ vs. $76.38\%$) over CLIP in new accuracy, respectively. Additionally, $DeKg_{KgCoOp}$ has a similar trend. This shows the benefits of DeKg by optimizing context explicit guidance by general and target knowledge, which aids base and new classes respectively.

PromptSRC and TCP are two strong competitors because they both leverage task-specific knowledge and general knowledge together to improve generalization. Fortunately, $DeKg_{TCP}$ demonstrates improved performance in recognizing both base and new classes. Specifically, $DeKg_{TCP}$ outperforms PromptSRC on 8 out of 11 datasets in terms of base accuracy and almost half of the datasets in new accuracy. Additionally, $DeKg_{TCP}$ shows improvement over TCP in almost all 11 datasets. The main reason is that PromptSRC and TCP guide the prompt with the token alignment strategy, limited by

Table 1: Comparison with existing methods in the base-to-new generalization setting with ViT-B/16 as the backbone. The context length $M$ is 4 for prompt-based methods with the 16-shot samples from the base classes. H: Harmonic mean.

| Datasets | | CLIP | CoOp | CoCoOp | MaPLe | KgCoOp | ProGrad | PromptSRC | TCP | DeKg$_{\text{KgCoOp}}$ | DeKg$_{\text{TCP}}$ |
|---|---|---|---|---|---|---|---|---|---|---|---|
| Regularization: | | | | | | consistency | | consistency and textual diversity | consistency and class-specific | consistency and independence | |
| Average | Base | 69.34 | 82.64 | 80.47 | 82.28 | 80.73 | 82.48 | 84.26 | 84.13 | 82.59 | **84.96** |
| | New | 74.22 | 68.00 | 71.69 | 75.14 | 73.60 | 70.75 | 76.10 | 75.36 | 74.93 | **76.38** |
| | H | 71.70 | 74.61 | 75.83 | 78.55 | 77.00 | 76.16 | 79.97 | 79.51 | 78.57 | **80.44** |
| ImageNet | Base | 72.43 | 76.46 | 75.98 | 76.66 | 75.83 | 77.02 | **77.60** | 77.27 | 76.65 | 77.40 |
| | New | 68.14 | 66.31 | 70.43 | 70.54 | 69.96 | 66.66 | 70.73 | 69.87 | 69.66 | 69.20 |
| | H | 70.22 | 71.02 | 73.10 | 73.47 | 72.78 | 71.46 | **74.01** | 73.38 | 72.99 | 73.07 |
| Caltech | Base | 96.84 | 98.11 | 97.96 | 97.74 | 97.72 | 98.02 | 98.10 | 98.23 | 98.13 | **98.64** |
| | New | 94.00 | 93.52 | 93.81 | 94.36 | 94.39 | 93.89 | 94.03 | 94.67 | 95.09 | 95.20 |
| | H | 95.40 | 95.76 | 95.84 | 96.02 | 96.03 | 95.91 | 96.02 | 96.42 | 96.59 | 96.89 |
| Pets | Base | 91.17 | 94.24 | 95.20 | **95.43** | 94.65 | 95.07 | 95.33 | 94.67 | 95.00 | 94.47 |
| | New | 97.26 | 96.66 | 97.69 | 97.76 | 97.76 | 97.63 | 97.30 | 97.20 | 97.71 | 97.76 |
| | H | 94.12 | 95.43 | 96.43 | 96.58 | 96.18 | 96.33 | 96.30 | 95.92 | 96.34 | 96.09 |
| Cars | Base | 63.37 | 76.20 | 70.49 | 72.94 | 71.76 | 77.68 | 78.27 | 80.80 | 76.31 | **81.18** |
| | New | 74.89 | 69.14 | 73.59 | 74.00 | 75.04 | 68.63 | 74.97 | 74.13 | 75.27 | 74.75 |
| | H | 68.65 | 72.50 | 72.01 | 73.47 | 73.36 | 72.88 | 76.58 | 77.32 | 75.79 | **77.83** |
| Flowers | Base | 72.08 | 97.63 | 94.87 | 95.92 | 95.00 | 95.54 | 98.07 | 97.73 | 97.72 | **98.58** |
| | New | **77.80** | 69.55 | 71.75 | 72.46 | 74.73 | 71.87 | 76.50 | 75.57 | 74.04 | 75.18 |
| | H | 74.83 | 81.23 | 81.71 | 82.56 | 83.65 | 82.03 | **85.95** | 85.23 | 84.25 | 85.30 |
| Food | Base | 90.10 | 89.44 | 90.70 | 90.71 | 90.50 | 90.37 | 90.67 | 90.57 | 90.57 | **90.73** |
| | New | 91.22 | 87.50 | 91.29 | **92.05** | 91.70 | 89.59 | 91.53 | 91.37 | 91.95 | 91.55 |
| | H | 90.66 | 88.46 | 90.99 | **91.38** | 91.09 | 89.98 | 91.10 | 90.97 | 91.25 | 91.14 |
| Aircraft | Base | 27.19 | 39.24 | 33.41 | 37.44 | 36.21 | 40.54 | 42.73 | 41.97 | 39.08 | **45.20** |
| | New | 36.29 | 30.49 | 23.71 | 35.61 | 33.55 | 27.57 | **37.87** | 34.43 | 34.97 | 35.09 |
| | H | 31.09 | 34.32 | 27.74 | 36.50 | 34.83 | 32.82 | **40.15** | 37.83 | 36.91 | 39.51 |
| SUN397 | Base | 69.36 | 80.85 | 79.74 | 80.82 | 80.29 | 81.26 | 82.67 | 82.63 | 81.19 | 82.52 |
| | New | 75.35 | 68.34 | 76.86 | **78.70** | 76.53 | 74.17 | 78.47 | 78.20 | 76.57 | 78.30 |
| | H | 72.23 | 74.07 | 78.27 | 79.75 | 78.36 | 77.55 | 80.52 | 80.35 | 78.81 | 80.35 |
| DTD | Base | 53.24 | 80.17 | 77.01 | 80.36 | 77.55 | 77.35 | 83.37 | 82.77 | 80.90 | **83.80** |
| | New | 59.90 | 47.54 | 56.00 | 59.18 | 54.99 | 52.35 | **62.97** | 58.07 | 58.21 | 59.66 |
| | H | 56.37 | 59.69 | 64.85 | 68.16 | 64.35 | 62.45 | **71.75** | 68.25 | 67.70 | 69.70 |
| EuroSAT | Base | 56.48 | 91.54 | 87.49 | **94.07** | 85.64 | 90.11 | 92.90 | 91.63 | 88.29 | 94.02 |
| | New | 64.05 | 54.44 | 60.04 | 73.23 | 64.34 | 60.89 | 73.90 | 74.73 | 72.69 | **81.69** |
| | H | 60.03 | 68.28 | 71.21 | 82.3 | 73.48 | 72.67 | 82.32 | 82.32 | 79.73 | **87.42** |
| UCF101 | Base | 70.53 | 85.14 | 82.33 | 83.00 | 82.89 | 84.33 | 87.10 | 87.13 | 84.64 | **88.06** |
| | New | 77.50 | 64.47 | 73.45 | 78.66 | 76.67 | 74.94 | 78.80 | 80.77 | 78.04 | **81.77** |
| | H | 73.85 | 73.38 | 77.67 | 80.77 | 79.65 | 79.35 | 82.74 | 83.83 | 81.21 | **84.80** |

handling domain shift in the test set. This demonstrates that DeKg$_{\text{TCP}}$ gains advantages by taking into account the textual embedding distribution with an independence constraint.

### 4.2.2 CROSS-DATASET GENERALIZATION

To further demonstrate that the proposed model can bridge the distribution gap between the pre-training dataset and the downstream evaluation set for zero-shot generalization, we compare DeKg with baselines under the cross-dataset generalization. In this experiment, we follow the baselines to regard ImageNet as the source dataset and the other 10 datasets as target datasets, i.e., there is a *distribution shift* between the base and new tasks.

From the comparison results in Table 2, we can see that our DeKg$_{\text{TCP}}$ obtains the highest average performance among all baselines (66.64% vs. 66.29% of TCP). By comparison, the performance on other datasets with distant and more fine-grained or specialized categories is much lower, such as Aircraft where the accuracy number is well below 30%. Nonetheless, DeKg$_{\text{TCP}}$ exhibits much stronger transferability than TCP with an average gain of 1.60% (i.e., 25.05% vs. 23.45%) on Aircraft, as well as most other fine-grained or specialized datasets. Additionally, DeKg$_{\text{KgCoOp}}$ achieves inferior performance to PromptSRC and TCP, mainly due to the inability to explicitly model the downstream class distribution.

### 4.2.3 FEW-SHOT CLASSIFICATION

To verify the model's ability to develop robust representations with a severely limited amount of downstream data, we follow the previous work (Yao et al., 2024) to train the model using $K$-shot labeled source images from each class and evaluate the testing domain with the same spaces as the training classes. A summary comparison of the 4-shot setting between the proposed DeKg and existing baselines appears in Table 3, from which we can observe that: the proposed DeKg$_{\text{TCP}}$ achieves the best average performance than all baselines. In addition, the baselines KgCoOp and TCP have

Table 2: Comparison in the cross-dataset generalization. The model is trained on the entire class of ImageNet (16 shots) and evaluated on the other 10 datasets.

| Datasets | CLIP | CoOp | CoCoOp | MaPLe | ProGrad | PromptSRC | KgCoOp | TCP | DeKg$_{KgCoOp}$ | DeKg$_{TCP}$ |
|---|---|---|---|---|---|---|---|---|---|---|
| ImageNet | 66.70 | 71.51 | 71.02 | 70.72 | 72.24 | 71.27 | 70.66 | 71.40 | 71.34 | **72.33** |
| Caltech101 | 93.30 | 93.70 | 94.43 | 93.53 | 91.52 | 93.60 | 93.92 | 93.97 | 93.87 | **94.73** |
| Pets | 89.10 | 89.14 | 90.14 | 90.49 | 89.64 | 90.25 | 89.83 | **91.25** | 90.16 | 90.02 |
| Cars | 65.70 | 64.51 | 65.32 | 65.57 | 62.39 | 65.70 | 65.41 | 64.69 | **65.91** | 65.49 |
| Flowers | 70.70 | 68.71 | 71.88 | 72.20 | 67.87 | 70.25 | 70.01 | 71.21 | 70.60 | **72.39** |
| Food101 | 85.90 | 85.30 | 86.06 | 86.20 | 85.40 | 86.15 | 86.36 | **86.69** | 86.37 | 86.59 |
| Aircraft | 24.90 | 18.47 | 22.94 | 24.74 | 20.16 | 23.90 | 22.51 | 23.45 | 23.37 | **25.05** |
| SUN397 | 62.60 | 64.15 | 67.36 | 67.01 | 62.47 | 67.10 | 66.16 | 67.15 | 66.11 | **67.19** |
| DTD | 44.30 | 41.92 | 45.73 | 46.49 | 39.42 | **46.87** | 46.35 | 44.35 | 46.16 | 44.47 |
| EuroSAT | 48.30 | 46.39 | 45.37 | 48.06 | 43.46 | 45.50 | 46.04 | **51.45** | 43.15 | 51.37 |
| UCF101 | 67.60 | 66.55 | 68.21 | 68.69 | 64.29 | 68.75 | 68.50 | 68.73 | 68.17 | **68.78** |
| Avg. | 65.24 | 63.88 | 65.74 | 66.30 | 62.71 | 65.81 | 65.51 | 66.29 | 65.33 | **66.64** |

Table 3: Comparison of few-shot learning with 4-shot samples.

| Datasets | CLIP | CoOp | CoCoOp | MaPLe | ProGrad | PromptSRC | KgCoOp | TCP | DeKg$_{KgCoOp}$ | DeKg$_{TCP}$ |
|---|---|---|---|---|---|---|---|---|---|---|
| ImageNet | 66.70 | 69.37 | 70.55 | 70.67 | 70.21 | **70.80** | 70.19 | 70.48 | 70.24 | 70.19 |
| Caltech101 | 93.30 | 94.44 | 94.98 | 94.30 | 94.93 | 94.77 | 94.65 | 95.00 | 94.97 | **95.21** |
| Pets | 89.10 | 91.30 | 93.01 | 92.05 | 93.21 | **93.23** | 93.20 | 91.90 | 93.10 | 92.15 |
| Cars | 65.70 | 72.73 | 69.10 | 68.70 | 71.75 | 71.83 | 71.98 | **76.30** | 72.24 | 74.90 |
| Flowers | 70.70 | 91.14 | 82.56 | 80.80 | 89.98 | 91.31 | 90.69 | 94.40 | 90.50 | **95.21** |
| Food101 | 85.90 | 82.58 | 86.64 | **86.90** | 85.77 | 86.06 | 86.59 | 85.30 | 86.88 | 85.72 |
| FGVC | 24.90 | 33.18 | 30.87 | 29.03 | 32.93 | 32.80 | 32.47 | 36.20 | 32.88 | **37.02** |
| SUN397 | 62.60 | 70.13 | 70.50 | 71.47 | 71.17 | 72.80 | 71.79 | 72.11 | 72.33 | **72.85** |
| DTD | 44.30 | 58.57 | 54.79 | 54.73 | 57.72 | 60.64 | 58.31 | 63.97 | 61.05 | **64.24** |
| EuroSAT | 48.30 | 68.62 | 63.83 | 54.87 | 70.84 | 75.02 | 71.06 | 77.43 | 72.65 | **79.16** |
| UCF101 | 67.60 | 77.41 | 74.99 | 73.70 | 77.82 | 79.35 | 78.40 | 80.83 | 79.43 | **81.05** |
| Avg. | 65.37 | 73.59 | 71.98 | 70.66 | 74.21 | 75.33 | 74.48 | 76.72 | 75.12 | **77.06** |

shown respective improvements with independence constraint (i.e., DeKg$_{KgCoOp}$ and DeKg$_{TCP}$) of the average gains of $0.64\%$ (i.e., $75.12\%$ vs. $74.48\%$) and $0.44\%$ (i.e., $77.06\%$ vs. $76.12\%$) across 11 datasets. This demonstrates that optimizing the learnable prompts with independence and consistency constraints together is indeed beneficial.

Next, we will conduct more detailed investigations for DeKg. If there is no special statement, all reported results represent average performance across over 11 datasets.

### 4.2.4  ABLATION STUDY AND ANALYSIS

To investigate the learning process of DeKg, we conduct ablative analysis including as follows:

**Effect of the Constraints Employed in DeKg**: DeKg contains two key constraints, including the consistency constraint $\mathcal{L}_{kg}$ and the independence constraint $\mathcal{L}_{kd}$. We conduct a constraint-wise analysis by adding one or two of them to the baseline method CoOp. Table 4 shows the results. We can see that the baseline CoOp provides high base class performance but suffers from poor generalization. By incorporating $\mathcal{L}_{kg}$ alone, the performance of new classes increases significantly by $5.62\%$ (i.e., $73.61\%$ vs. $67.99\%$), but the base classes degrades from $82.63\%$ to $80.73\%$. This suggests that $\mathcal{L}_{kg}$ explicitly enforces the intra-class consistency between the learnable and crafted prompts to capture generalizable features from frozen CLIP. In contrast, incorporating $\mathcal{L}_{kd}$ alone leads improvements in both base and new classes compared with CoOp, indicating its ability in balancing model adaptation and generalization. It achieves the best performance in base classes but still lags behind KgCoOp in new classes. The main reason lies in that $\mathcal{L}_{kd}$ aims to capture the task-specific knowledge by enhancing the divergence among learnable prompts across various classes with prior inter-class relationships inherent in the crafted prompts. Finally, combining $\mathcal{L}_{kd}$ and $\mathcal{L}_{kg}$, DeKg achieves improvements in both base and new classes, leading to the average new class and harmonic mean gains of $6.94\%$ (i.e., $74.93\%$ vs. $67.99\%$) and $3.97\%$ (i.e., $78.57\%$ vs. $74.60\%$). It is reasonable because $L_{kg}$ and $L_{kd}$ are complementary, one for intra-class consistency and the other for inter-class divergence.

**Comparison of Model Complexity**: To better understand the benefits of the proposed DeKg, we examined the model complexity. As shown in Table 5, it can be observed that DeKg is an efficient

Table 4: Effect of the constraints in our model.

| Method | Base | New | H |
|--------|------|-----|---|
| CoOp | 82.63 | 67.99 | 74.60 |
| $+\mathcal{L}_{kg}$ | 80.73 | 73.61 | 77.00 |
| $+\mathcal{L}_{kd}$ | **83.13** | 69.87 | 75.93 |
| $+\mathcal{L}_{kg} + \mathcal{L}_{kd}$ (ours) | 82.59 | **74.93** | **78.57** |

Table 5: Comparison of model complexity with KgCoOp.

| Method | Total Parameters (M) | H |
|--------|---------------------|---|
| KgCoOp | 124.325 | 77.00 |
| DeKg$_{\text{KgCoOp}}$ | +0 | 78.57 |
| TCP | +0.329 | 79.51 |
| DeKg$_{\text{TCP}}$ | +0.329 | **80.44** |

method that performs better with the same model complexity of corresponding baselines. For example, DeKg$_{\text{KgCoOp}}$ shows an average improvement of $1.57\%$ (i.e., $78.57\%$ vs. $77.00\%$) over KgCoOp without adding any parameters. The main reason is that DeKg simply adds an efficient regularization for generating discriminative classifiers guided by better knowledge, i.e., optimizing the balance between adapting general knowledge and fine-tuning for targeted tasks.

**Effect of Hyperparameter $\lambda$ and $\mu$**: To further investigate the impact of consistency and independence constraints on model performance, we analyze the effect of hyperparameter $\lambda$ and $\mu$ in the proposed model DeKg, i.e., $\lambda$ controls the contribution of capturing general knowledge, and $\mu$ controls the divergence between task-specific knowledge and general knowledge. The effect of $\lambda$ and $\mu$ on DeKg with KgCoOp and TCP (i.e., DeKg$_{\text{KgCoOp}}$ and DeKg$_{\text{TCP}}$) is shown in Figure 3a and Figure 3b, respectively. It can be seen that the performance of new tasks becomes better as $\lambda/\mu$ increases, indicating that the consistency constraint effectively captures the essential knowledge for new classes. The results reach the best when $\lambda/\mu = 3/1$ for both DeKg$_{\text{KgCoOp}}$ and DeKg$_{\text{TCP}}$. After that, the performance decreases because a larger ratio forces the learnable prompts to rely strongly on general knowledge, failing to capture task-specific information. The trend for base precision is reversed. This result is reasonable because a larger ratio of $\lambda/\mu$ reduces the importance of task-specific knowledge, which is essential for base tasks.

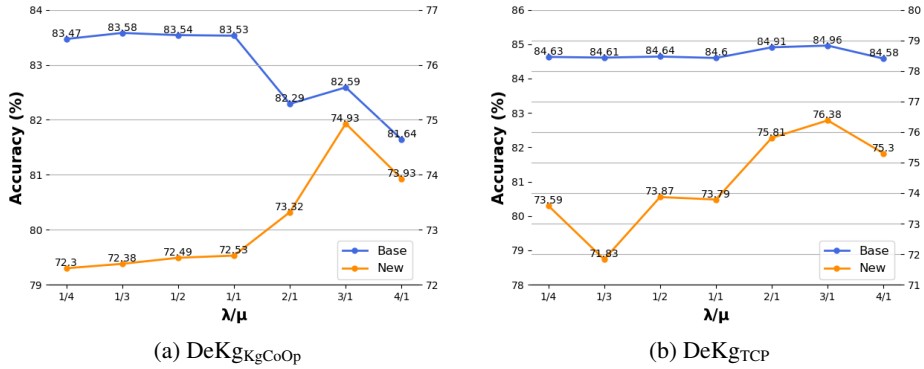

(a) DeKg$_{\text{KgCoOp}}$        (b) DeKg$_{\text{TCP}}$

Figure 3: Effect of hyperparameters $\lambda$ and $\mu$ on DeKg$_{\text{KgCoOp}}$ and DeKg$_{\text{TCP}}$.

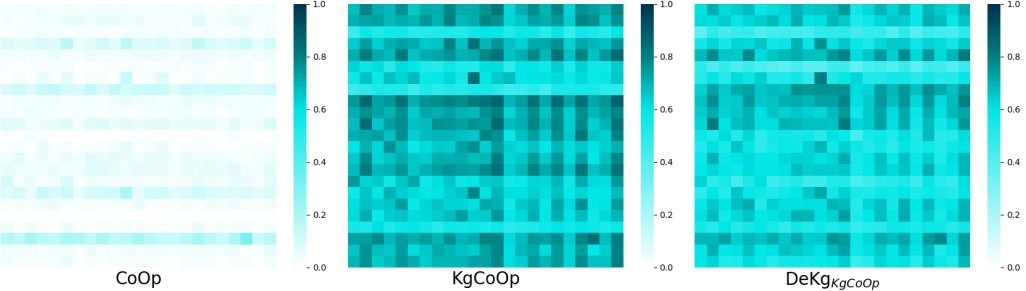

Figure 4: Visualization the HSIC values between $\mathbf{W}$ and $\mathbf{W}^{clip}$ in DTD dataset.

**Visualization**: As shown in Figure 4, the HSIC values obtained from the consistency-constrained method KgCoOp are very high. This indicates that the learnable tokens are highly correlated with

the pre-trained general knowledge, which can lead to poor performance on target tasks. In contrast, the HSIC values derived without knowledge-guided context optimization method CoOp are very low. This suggests a weak reliance on general knowledge and a tendency to overfit the target task, resulting in limited generalization ability for target tasks. The values obtained by DeKg are moderate compared to the baselines. This suggests that the proposed HSIC regularization term $\mathcal{L}_{kd}$ effectively maintains a balance between dependence on general knowledge and task-specific knowledge.

## 5 CONCLUSION

Knowledge-guided context optimization is a representative visual-language prompt tuning framework. It emphasizes the consistency between the learnable and crafted prompts to alleviate catastrophic forgetting, which boosts the generalization ability but degrades the few-shot learning ability in downstream tasks. In this paper, we propose a DeKg strategy by introducing an independence constraint, which exploits Hilbert-Schmidt Independence Criterion regularization to enhance the divergence between learnable and crafted prompt for capturing task-specific unique knowledge, thereby enhancing the model's discriminative capability. Extensive evaluations on three challenging benchmarks demonstrate that DeKg is an effective and efficient prompt tuning method. Specifically, one of the main advantages of DeKg is its ability to seamlessly integrate with existing knowledge-guided context optimization methods, such as KgCoOp and TCP, significantly enhancing their performance without requiring additional parameters. Another key advantage of DeKg is its capability to outperform strong baselines on both base classes and new classes in base-to-new settings, while existing methods often struggle to keep a balance between the generalization ability and few-shot learning ability. Additionally, it demonstrates superior performance in cross-dataset generation and few-shot learning scenarios. In the future, we plan to incorporate DeKg to more visual-language prompt tuning frameworks and applications.

ACKNOWLEDGEMENTS

This work is supported by grants from the National Natural Science Foundation of China (Grant Nos. 62306188, 62376166) and the R&D Program of the Beijing Municipal Education Commission (KM202410028013). The Center for Experimental Teaching of Computer Science & Technology at Capital Normal University provides computational support.

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
