# OpenReview forum: "Divergence-enhanced Knowledge-guided Context Optimization for Visual-Language Prompt Tuning"
_ICLR.cc/2025/Conference — ICLR 2025 Poster_

### Official Review · Reviewer_nZxi · 2024-10-31

**Soundness:** 3
**Presentation:** 2
**Contribution:** 1
**Rating:** 5
**Confidence:** 4

**Summary:**

This paper proposes to adapt HSIC as an extra regularization term, which achieves a better trade-off between the performance on base and new classes. The experiments show the effectiveness of this regularization on varies experiment settings. The paper is well written and easy to understand.

**Strengths:**

1. I think this paper has a reasonable motivation to maximize the independence between learnable prompt and manual prompt.
2. This paper has a very extensive experiment analysis on varies clip model adaptation task and show good results.

**Weaknesses:**

1. More analysis is needed to discuss why HISC is chosen as the metrics to measure the prompt independence. Other methods like information bottleneck can do that too.
2. L_kd and L_kg seems to be a pair of totally contradictive losses. I wonder if this will cause the model to be difficult to converge. It would be better to provide more analysis on how the loss weight of these two losses affect the model convergence.
3. More performance comparison and analysis on other state-of-the-art prompt tuning method like:
Yubin, et.al, Learning Hierarchical Prompt with Structured Linguistic Knowledge for Vision-Language Models

**Questions:**

Please refer to the concern in the Weakness section.

---

> ### Author Response · Authors · 2024-11-22
> **Responce to Reviewer nZxi (1/2)**
>
> Thank you for your encouraging and helpful suggestions. Below, we address the comments you provided.
>
>  **Q1.Why was HSIC chosen as the metric for prompt independence? Could other methods like information bottleneck be used instead?**
>
>
> In optimizing the balance between adapting general knowledge and fine-tuning for target tasks, it is essential to measure the independence between learnable and crafted prompts. With the benefit of non-parametrics, easy computability, rapid convergence, and small estimation bias with finite samples (Ma W D K, Lewis J P, Kleijn W B. The HSIC bottleneck: Deep learning without back-propagation,Proceedings of the AAAI conference on artificial intelligence. 2020, 34(04): 5085-5092.), the HSIC is employed to encourage the learnable prompts to maintain a consistent yet independent relation with general knowledge.
>
> To verify the effectiveness of using HSIC as an independence measure, we implement an additional experiment incorporating Mutual Information (MI) as the independence constraint with the knowledge-guided context optimization method KgCoOp under the base-to-new generalization setting. The MI loss is formulated as $\mathcal L_{MI}=H(\mathbf W)+H(\mathbf W^{clip}-H(\mathbf W, W^{clip})$, where $H$ is the function of entropy. Compared with MI, HSIC avoids the complex probability density estimation under high-dimensional data through the kernel method, which is more efficient and more robust to dimension expansion. The comparison results are summarized below.
>
> Table 1:Comparison of different independence constraints used in DeKg, where MI is the mutual information constraint.
> | Dataset      | KgCoOp |       |       | DeKg with HSIC |       |       | DeKg with MI |        |       |
> |--------------|:------:|:-----:|:-----:|:--------------:|:-----:|:-----:|:------------:|:------:|:-----:|
> |              |  Base  |  New  |   H   |      Base      |  New  |   H   |     Base     |   New  |   H   |
> | ImageNet     |  75.83 | 69.96 | 72.78 |      76.65     | 69.66 | 72.99 |     75.84    |  69.65 | 72.61 |
> | Caltech101   |  97.72 | 94.39 | 96.03 |      98.13     | 95.09 | 96.59 |     97.85    |  94.50  | 96.15 |
> | OxfordPets   |  94.65 | 97.76 | 96.18 |       95.00       | 97.71 | 96.34 |     94.74    |  97.71 |  96.20 |
> | StandfordCar |  71.76 | 75.04 | 73.36 |      76.31     | 75.27 | 75.79 |     73.30     |  74.80  | 74.04 |
> | Flowers      |   95.00   | 74.73 | 83.65 |      97.72     | 74.04 | 84.25 |     95.79    |  74.85 | 84.04 |
> | Food101      |  90.50  |  91.70 | 91.09 |      90.57     | 91.95 | 91.25 |     90.06    |  91.72 | 90.88 |
> | FGVCAircraft |  36.21 | 33.55 | 34.83 |      39.08     | 34.97 | 36.91 |     38.08    |  32.99 | 35.35 |
> | SUN397       |  80.29 | 76.53 | 78.36 |      81.19     | 76.57 | 78.81 |     83.83    |  74.92 | 79.12 |
> | DTD          |  77.55 | 54.99 | 64.35 |      80.90      | 58.21 |  67.7 |     79.24    |  57.05 | 66.34 |
> | EuroSAT      |  85.64 | 64.34 | 73.48 |      88.29     | 72.69 | 79.73 |     86.60     |  63.79 | 73.47 |
> | UCF101       |  82.89 | 76.67 | 79.65 |      84.64     | 78.04 | 81.21 |     80.92    |  76.22 |  78.50 |
> | Avg.         |  80.73 | 73.61 |   77.00  |     82.59     | 74.93 | 78.57 |     81.48    | 73.47 | 77.27 |
>
> It can be seen that our method constrained with HSIC consistently obtains the best average performance across 11 datasets. Specifically, the proposed method with HSIC has shown respective improvement of the average gains of 1.11% (i.e., 82.59% vs 81.48%) on base accuracy, 1.46% (i.e., 74.93% vs 73.47%) on new accuracy, and 1.30% (i.e., 78.57% vs 77.27%), respectively. This demonstrates that constrained independence using HSIC is indeed beneficial.
>
> **Q2.$\mathcal L_{kd}$ and $\mathcal L_{kg}$ seems to be a pair of totally contradictive losses. Does the interaction between $\mathcal L_{kd} $ and $\mathcal L_{kg} $ cause convergence issues?**
>
> $
> \mathcal L_{kg}=||\mathbf W-\mathbf W^{clip}||_2^2,
> $
>
> $
> \mathcal L_{kd}=(N_c-1)^{-2}tr(\mathbf K \mathbf H \mathbf K^{clip} \mathbf H).
> $
>
> The consistency constraint $\mathcal L_{kg}$ and the independency constraint $\mathcal L_{kd}$ are both convex functions, enabling the existence of an optimal solution for the variable $\mathbf W$ and the convergence of the final objective function. To further investigate the model convergence, the convergence curves of KgCoOp (i.e., $ \mathcal L_{ce}+\lambda\mathcal L_{kg}$) and DeKg   (i.e., $\mathcal L_{ce}+\lambda\mathcal L_{kg}+\mu \mathcal L_{kd}$) are shown in the Figure 1 in the Supporting Material Appendix . It can be seen that the objective function values of both  KgCoOp and  DeKg$_\text{KgCoOp}$ are stable after 80 epochs.

---

> ### Author Response · Authors · 2024-11-22
> **Responce to Reviewer nZxi (2/2)**
>
> **Q3. Comparison with other state-of-the-art methods (e.g., Yubin et al., Learning Hierarchical Prompt with Structured Linguistic Knowledge for Vision-Language Models)**
>
> Paper ``Yubin et al. Learning Hierarchical Prompt with Structured Linguistic Knowledge for Vision-Language Models '' is an important work. We will cite it. As suggested by the reviewer, We added the experiment results of HPT (Yubin et al., Learning Hierarchical Prompt with Structured Linguistic Knowledge for Vision-Language Models) in Table 2.
>
> From the comparison shown below, it can be seen that $DeKg_\text{TCP}$ and $DeKg_\text{PromptSRC}$ almost perform as well as HPT across 11 datasets on base-to-new generalization. Compared to HPT which incorporates both structured and conventional linguistic knowledge from LLMs for enhancing prompt effectiveness in a hierarchical manner, our proposed DeKg approach only integrates independence constraint $\mathcal L_{kd}$ into existing knowledge-guided contextual optimization methods (i.e., KgCoOp, TCP, and PromptSRC) without the help of any external information, which is simple and effective. In addition, It can serve as a plug-and-play module to boost the performance of existing knowledge-guided methods.
>
> Table 2: Comparison of HPT and our methods on the base-to-new generalization.
> | Datasets |      |  HPT  | DeKg$_\text{KgCoOp}$ | DeKg$_\text{TCP}$ | DeKg$_\text{PromptSRC}$ |
> |:--------:|:----:|:-----:|:--------------------:|:------------:|:------------------:|
> |  Average | Base | 84.32 |         82.59        |     84.96    |        84.22       |
> |          |  New | 76.86 |         74.93        |     76.38    |        76.28       |
> |          |   H  | 80.23 |         78.57        |     80.44    |        80.06       |
> | ImageNet | Base | 77.95 |         76.65        |     77.40    |        77.60        |
> |          |  New | 70.74 |         69.66        |     69.20    |        70.52       |
> |          |   H  | 74.17 |         72.99        |     73.07    |        73.89       |
> |  Caltech | Base | 98.37 |         98.13        |     98.64    |        98.17       |
> |          |  New | 94.98 |         95.09        |     95.20    |        93.82       |
> |          |   H  | 96.65 |         96.59        |     96.89    |        95.95       |
> |   Pets   | Base | 95.78 |         95.00        |     94.47    |        95.13       |
> |          |  New | 97.65 |         97.71        |     97.76    |        96.76       |
> |          |   H  | 96.71 |         96.34        |     96.09    |        95.94       |
> |   Cars   | Base | 76.95 |         76.31        |     81.18    |        78.03       |
> |          |  New | 74.23 |         75.27        |     74.75    |        75.55       |
> |          |   H  | 75.57 |         75.79        |     77.83    |        76.77       |
> |  Flowers | Base | 98.17 |         97.72        |     98.58    |        97.63       |
> |          |  New | 78.37 |         74.04        |     75.18    |        77.49       |
> |          |   H  | 87.16 |         84.25        |     85.30    |        86.40        |
> |   Food   | Base | 90.46 |         90.57        |     90.73    |        90.75       |
> |          |  New | 91.57 |         91.95        |     91.55    |        91.51       |
> |          |   H  | 91.01 |         91.25        |     91.14    |        91.13       |
> | Aircraft | Base | 42.68 |         39.08        |     45.20    |        42.58       |
> |          |  New | 38.13 |         34.97        |     35.09    |        37.55       |
> |          |   H  | 40.28 |         36.91        |     39.51    |        39.91       |
> |  SUN397  | Base | 82.57 |         81.19        |     82.52    |        82.59       |
> |          |  New | 79.26 |         76.57        |     78.30    |        78.84       |
> |          |   H  | 80.88 |         78.81        |     80.35    |        80.67       |
> |    DTD   | Base | 83.84 |         80.90        |     83.80    |        83.53       |
> |          |  New | 63.33 |         58.21        |     59.66    |         63.00         |
> |          |   H  | 72.16 |         67.70        |     69.70    |        71.83       |
> |  EuroSAT | Base | 94.24 |         88.29        |     94.02    |        93.38       |
> |          |  New | 77.12 |         72.69        |     81.69    |        75.36       |
> |          |   H  | 84.82 |         79.73        |     87.42    |        83.41       |
> |  UCF101  | Base | 86.52 |         84.64        |     88.06    |        87.02       |
> |          |  New | 80.06 |         78.04        |     81.77    |        78.73       |
> |          |   H  | 83.16 |         81.21        |     84.80    |        82.67       |
>
> Thank you again for your valuable feedback. If you have any additional questions or suggestions, we would be happy to address them.

---

### Official Review · Reviewer_z32d · 2024-11-01

**Soundness:** 3
**Presentation:** 3
**Contribution:** 2
**Rating:** 6
**Confidence:** 3

**Summary:**

This paper proposes a novel method called Divergence-enhanced Knowledge-guided Prompt Tuning (DeKg), which employs Hilbert-Schmidt Independence Criterion (HSIC) regularization to maintain a degree of independence between the learnable prompts and pre-trained knowledge, addressing the bias problem caused by over-reliance on pre-trained knowledge. Built upon knowledge-guided context optimization, DeKg introduces an independence constraint, enabling learnable prompts to retain consistency with general knowledge while capturing task-specific features, thus achieving a better balance between base and novel classes.

**Strengths:**

1. The paper addresses the inherent bias issue in knowledge-guided context optimization by introducing a novel Hilbert-Schmidt Independence Criterion (HSIC)-based regularization that encourages independence between learnable and crafted prompts.
2. DeKg integrates with existing methods, enhancing class-specific prompt distinction without increasing model complexity.

**Weaknesses:**

1. The motivation of using HSIC as the constrain is not clearly elaborated. Further analysis of your motivation will be insightful.
2. One of the proposed loss: $L_{kg}$ is already applied in existing methods, such as KgCoOp and PromptSRC, which weakens the novelty of overall method.

**Questions:**

1. As mentioned in weakness1, more analysis of your motivation concerning why you choose Hilbert-Schmidt Independence Criterion would be insightful.
2. The experiment of domain generalization seems to be missed in your paper, which is conducted by most prompt tuning method. Could you provide Dekg’s performance on this setting.
3. In Tabel 4, it is obvious that the $L_{kg}$ works well in novel, while $L_{kd}$ performs well in base. Dose that mean these two losses are strongly coupled and your proposed $L_{kd}$ is not recommended to use independently? More analysis on above question would be insightful.

---

> ### Author Response · Authors · 2024-11-22
> **Responce to Reviewer z32d (1/2)**
>
> Thank you for your encouraging and helpful suggestions. Below, we address the comments you provided.
>
> ---
>
> **Q1. Why was HSIC chosen as the constraint, and what is the specific motivation behind using the Hilbert-Schmidt Independence Criterion?**
>
> We reorganize subsection 3.2 to clarify the motivation, i.e., the contradiction problem between catastrophic forgetting in fine-tuning and bias in pre-training.
>
> Despite the consistency between learnable tokens and general knowledge playing an important role in preventing catastrophic forgetting, it still encounters substantial challenges, i.e., the bias towards pre-trained models. Due to the inherent differences between fine-tuning and pre-training, particularly when the data distribution of the target task differs from that of the pre-training data, the learnable tokens optimized with limited downstream trainable data will inevitably lean towards the distributions of pre-trained knowledge, resulting in the performance degradation on the target tasks.
>
> Compared to matching learnable and crafted tokens, the relationship between classes from the frozen general knowledge can be more informative for the target task. Therefore, we consider transferring pairwise relevance from general knowledge to the target task. Concretely, given the embedding of an anchor class $j$, the pairwise relevance over all classes can be computed as $\mathbf K(i,j )=k(\mathbf w_i,\mathbf w_j)$, $\mathbf K(i,j )^{clip}=k(\mathbf w_i^{clip},\mathbf w_j^{clip})$, where $\mathbf w_i$ denotes the learnable tokens of class $i$, and the corresponding crafted tokens is $\mathbf w_i^{clip}$, $k(\cdot, \cdot)$ represents the kernel function, in which the inner product kernel function can be adopted as $k(\mathbf w_i,\mathbf w_j)=\mathbf w_i^T \mathbf w_j$.
>
> Considering that the Hilbert-Schmidt Independence Criterion (HSIC) is widely used as an independence measurement with the benefit of non-parametric, easy computability, rapid convergence, and small estimation bias with finite samples (Ma W D K, Lewis J P, Kleijn W B. The HSIC bottleneck: Deep learning without back-propagation, Proceedings of the AAAI conference on artificial intelligence. 2020, 34(04): 5085-5092.), the independence between learnable and crafted prompts can be constrained with pair-wise relevance, which can be formulated as follows:
>
> $$
> \mathcal L_{kd}=HSIC(\mathbf W, \mathbf W^{clip})
>                 =(N_c-1)^{-2}tr(\mathbf K \mathbf H \mathbf K^{clip} \mathbf H)
>                 = (N_c-1)^{-2}\sum_{i,j} \mathbf K(i,j) \mathbf A_{i,j},
> $$
> where $N_c$ is the number of classes, and $\mathbf H=\mathbf I_{N_c}-\frac{1}{N_c}\mathbf 1_{N_c} \mathbf 1_{N_c}^T$ is the centering matrix. Specifically, the intra-class relevance can be formulated as $\mathcal L_{kd} (\mathbf w_i,\mathbf w_i^{clip})= (N_c-1)^{-2}\sum_{j} \mathbf K(i,j) \mathbf A_{j,i}$, and the inter-class relevance as $\mathcal L_{kd} (\mathbf w_i,\mathbf w_j^{clip})= (N_c-1)^{-2}\sum_{l} \mathbf K(i,l) \mathbf A_{l,j}$, Therefore, penalizing $\mathcal L_{kd}$ encourages both intra-class and inter-class independence to eliminate the bias towards pre-training.
>
> **Q2.One of the proposed loss: $\mathcal L_{kg}$is already applied in existing methods, such as KgCoOp and PromptSRC, which weakens the novelty of overall method.**
>
> We do not claim the attribution of proposing the $\mathcal L_{kg}$ constraint. Our work aims to overcome the weakness and limitations of $\mathcal L_{kg}$ and shows that integrating them together could obtain superior performance.

---

> ### Author Response · Authors · 2024-11-22
> **Responce to Reviewer z32d (2/2)**
>
> **Q3.Could you provide DeKG's performance on domain generalization, as this experiment is commonly included in prompt tuning methods?**
>
> We implement the experiment under the domain generalization setting. In this experiment, we follow the baselines to conduct the prompt tuning on the few-shot ImageNet, and evaluate the model on the ImangeNetV2, ImageNet-Sketch, ImageNet-A, and ImageNet-R datasets, i.e., there is a distribution shift within the same class. The related results are summarized below.
>
>
> Table 1:Comparison of domain generalization from ImageNet to its variants.
> |         | Source    | Target     | Target          | Target     | Target     |      |
> |---------|-----------|------------|-----------------|------------|------------|-----------|
> |         | ImageNet  | ImageNetV2 | ImageNet-Sketch | ImageNet-A | ImageNet-R | Avg.      |
> | CLIP    | 66.73     | 60.83      | 46.15           | 47.77      | 73.96      | 57.17     |
> | CoOp    | 71.51     | 64.20       | 47.99           | 49.71      | 75.21      | 59.28     |
> | CoCoOp  | 71.02     | 64.07      | 48.75           | 50.63      | 76.18      | 59.90      |
> | KgCoOp  | 71.20      | 64.10       | 48.97           | 50.69      | 76.7       | 60.11     |
> | DeKg$_\text{KgCoOp}$ | 71.34     | 64.12      | 48.92           | 50.37      | 76.62      | 60.01   |
> | TCP     | 71.40      | 64.50   | 49.53      | 51.10   | 76.73  | 60.51 |
> | DeKg$_\text{TCP}$    | 72.33 | 64.31      | 48.38           | 50.51      | 76.37      | 59.89   |
>
>
> As indicated by the comparison results, the proposed method performs slightly worse than or equal to the corresponding knowledge-guided context optimization methods. The primary reason is that the independence constraint mainly models the distribution of classes, but it cannot capture the target-specific instance-level distribution. In the future, we will consider instance-level distribution to capture the variance in real data and enhance our work.
>
>
> **Q4. Dose that mean these two losses are strongly coupled and your proposed $\mathcal L_{kd}$ is not recommended to use independently?**
>
> Indeed, these two distinct constraints are complementary, i.e., $\mathcal L_{kg}$ for preserving the pre-trained general knowledge to boost the performance in new tasks, and $\mathcal L_{kd}$ for capturing the task-specific knowledge in fine-tuning to enhance the performance in base tasks. The proposed method integrates two constraints to address the contradiction problem between catastrophic forgetting during fine-tuning and bias in pre-training, achieving the best overall performance. However, only utilizing one of them achieves suboptimal performance. The experiment results and discussions are given in Effect of constraints employed in DeKg of Subsection 4.2.
>
>
> Thank you again for your valuable feedback. If you have any additional questions or suggestions, we would be happy to address them.

---

### Official Review · Reviewer_Km4y · 2024-11-03

**Soundness:** 4
**Presentation:** 3
**Contribution:** 3
**Rating:** 6
**Confidence:** 5

**Summary:**

This paper tackles the inherent issue of knowledge-guided context optimization, which overly biases general knowledge in pre-training. It proposes a novel HISC-based regularization method, DeKg, for encouraging independence between the learnable and the crafted prompts. Extensive experiments demonstrate the superiority of the proposed method in three challenging benchmarks:

**Strengths:**

+Using the Hilbert-Schmidt Independence Criterion (HSIC) is an interesting topic for encouraging independence between learnable and crafted prompts, which can boost performance in the seen classes.

+Evaluation shows the effectiveness of the proposed method.

+The proposed DeKg integrates seamlessly with existing knowledge-guided methods.

**Weaknesses:**

-As shown in Figure 1, the proposed DeKg obtains a higher performance than the performance of CoOp for base classes and the zero-shot CLIP for new classes. However, the Hilbert-Schmidt Independence Criterion (HSIC) contained in DeKg is a constraint between the learnable and crafted prompts without injecting additional information. Why can the proposed DeKg obtain a better performance?

-L221: The proposed L_{kd} involves two terms: intra-class relations and inter-class relations. Moreover, the author claims that  penalizing
L_{kd} encourages both intra-class and inter-class independence. Furthermore, the intra-class consistency is formulated between w_i and w_{i}^{clip}, which is the same as the L_{kg}. In other words, the proposed HSIC has contained the knowledge consistency L_{kg}. Therefore, the final objective of Eq.(5) should not contain L_{kg} because L_{kd} has been constrained by the intra-class consistency. However, the results in Table 4 are inconclusive with the above conclusion. Even more unfortunate, L_{kd} performs worse than L_{kg}. Why?

-It is recommended to provide a code.

-Since the proposed HSIC is model-independent, it is suggested that the module's generalization and plug-and-play be verified using more CoOp-based methods.

**Questions:**

Please see #Weaknesses

---

> ### Author Response · Authors · 2024-11-22
> **Responce to Reviewer Km4y (1/2)**
>
> Thank you for your encouraging and helpful suggestions. Below, we address the comments you provided.
>
> ---
> **Q1.Why can the proposed DeKg obtain better performance despite HSIC not injecting additional information?**
>
> To ensure that learnable prompts retain essential general knowledge contained in frozen CLIP, existing knowledge-guided context optimization (KGCO) methods like KgCoOp and TCP emphasize the consistency between the Learnable and crafted prompts to alleviate catastrophic forgetting, which boosts the generalization ability but restricts the ability to capture task-specific knowledge, resulting in performance degradation in base tasks. To maintain the advantage of KGCO while allowing the adaptability to capture tasks-specific knowledge, we inject the Hilbert-Schmidt Independence Criterion (HSIC) regularization term into recent KGCO methods. This strategy guides context optimization with divergence-enhanced knowledge (DeKg), i.e., incorporating consistency and diversity to enhance the generalization and discriminative capabilities of the learnable tokens, thereby improving the performance both in base and new tasks. The proposed DeKg achieves better performance than existing KGCO methods without adding extra information.
>
>  **Q2. The proposed $\mathcal L_{kd}$ involves two terms: intra-class relations and inter-class relations. The intra-class consistency is formulated between $\mathbf w_i$ and $\mathbf w_{i}^{clip}$ in $\mathcal L_{kd}$, which is the same as the $\mathcal L_{kg}$. The final objective of Eq.(5) should not contain $\mathcal L_{kg}$ because $\mathcal L_{kd}$  has been constrained by the intra-class consistency. However, the results in Table 4 are inconclusive with the above conclusion. Even more unfortunate, $\mathcal L_{kd}$ performs worse than $\mathcal L_{kg}$. Why?**
>
> Firstly, the intra-class relations contained in independence constraint $\mathcal L_{kd}$ are different from the consistency constraint $\mathcal L_{kg}$. Specifically, $\mathcal L_{kg}$ enforces the feature representations obtained by learnable prompts to be consistent with the pre-trained CLIP features within the textual embedding space, i.e., $||\mathbf{w}_i - \mathbf{w}_i^{\text{clip}}||_2^2$ .
>
> However, the intra-class relations of $\mathcal L_{kd}$  enforces the pairwise similarity between learnable prompts to be consistent with the pre-trained CLIP features' pairwise similarity, i.e., $\mathcal L_{kd} (\mathbf w_i,\mathbf w_i^{clip})=\sum_j k(\mathbf w_i,\mathbf w_j) \mathbf H_j k(\mathbf w^{clip}_j,\mathbf w^{clip}_i)\mathbf H_i$, where $k(\cdot,\cdot)$ is a kernel function, and $\mathbf H$ is the centering matrix.
>
> That means $\mathcal L_{kg}$ aims to preserve the general knowledge while $\mathcal L_{kd}$ allows the divergence between the learnable and crafted prompts. Thus, $\mathcal L_{kg}$ plays an important role in final context optimization.
>
> Compared to solely using $\mathcal L_{kg}$, $\mathcal L_{kd}$ performs worse in new classes but better in base classes. The primary reason is that the learnable context inevitably overfits task-specific knowledge distributions by optimizing with the downstream trainable data without retaining the general knowledge. This limitation reduces its generalization ability and ultimately decreases overall performance. These constraints complement each other and achieve the best overall performance when integrated, the results in Table 4 are consistent with the above conclusions.
>
> **Q3. It is recommended to provide a code.**
>
> We guarantee that the source code will be public in the Github platform after this paper is accepted.

---

> > ### Comment · Reviewer_Km4y · 2024-11-26
> > **I still have several doubts about the response.**
> >
> > Thanks for your response which addresses most of my concerns. However, I still have several doubts about the response.
> >
> > Q1、“incorporating consistency and diversity to enhance the generalization and discriminative capabilities of the learnable tokens, thereby improving the performance both in base and new tasks”， Why incorporate the consistency and diversity can enhance the generalization and discriminative capabilities of the learnable tokens? This one needs to be explained in more detail.
> >
> > Q2、My main question is that the proposed objective function $L_{kd}$ already contains the intra-class constraints that have similar functions to $L_{kg}$, so why use the $L_{kg}$in the final objective function

---

> > > ### Author Response · Authors · 2024-11-27
> > > **Responce to Reviewer Km4y**
> > >
> > > Thank you for your positive feedback.
> > >
> > > **Q1:''incorporating consistency and diversity to enhance the generation and discriminative capabilities of the learnable tokens, thereby improving the performance both in base and new tasks.'' Why incorporate the consistency and diversity can enhance the generalization and discriminative capabilities of the learnable tokens? This one needs to be explained in more detail.**
> > >
> > > Given CLIP's remarkable zero-shot generalization performance, learnable tokens are expected to approximate crafted tokens by utilizing the consistency constraint $\mathcal L_{kg}$. While this approach improves performance on new tasks, it negatively impacts performance on base tasks. To address the bias towards pre-training, we propose the independence constraint $\mathcal{L}_{kd}$ to enhance the divergence between learnable and crafted tokens, capturing the discriminative task-specific knowledge and enhancing performance on base tasks. Consequently, the consistency and independence constraints complement each other, and ultimately enhance overall performance when incorporated into the final objective.
> > >
> > > From the comparison results shown in Table 4, it can be observed that, compared to CoOp, solely using $\mathcal L_{kg}$ or $\mathcal L_{kd}$ either just boosts new accuracy or base accuracy. However, integrating them yields the best overall performance. This indicates that optimizing the learnable prompts with both consistency and independence constraints together is indeed beneficial.
> > >
> > > **Q2:My main question is that the proposed objective function $\mathcal L_{kd}$ already contains the intra-class contains that that have similar functions to $\mathcal L_{kd}$, so why use the $\mathcal L_{kg}$ in the final objective.**
> > >
> > > $\mathcal L_{kd}$ is merely a regularization term for simultaneously constraining both intra-class and inter-class independence between learnable and crafted tokens, where the intra-class independence can be formulated as
> > >
> > > $$
> > > \mathcal{L}_{kd} (\mathbf{w}_i, \mathbf{w}_i^{clip}) = \underbrace{\sum_j \underbrace{k(\mathbf{w}_i, \mathbf{w}_j)}\_{\text{inter-class relevance}} \mathbf{H}_j \underbrace{k(\mathbf{w}_j^{clip}, \mathbf{w}_i^{clip})}\_{\text{inter-class relevance}} \mathbf{H}_i}\_{\text{intra-class relevance}}.
> > > $$
> > >
> > > Obviously, the intra-class independence constrained in $\mathcal L_{kd}$ is entirely different from $\mathcal L_{kg}$ (i.e., $||w_i-w_i^{clip}||_2^2$). Specifically, $\mathcal {L_kd}$ penalizes the intra-class relevance by considering the inter-class relevance of learnable and crafted tokens, but doesn't directly constrain intra-class relevance like $\mathcal {L_kg}$. Therefore, they do not have a containment relationship but complement each other, and both of them should be included in the final objective.

---

> ### Author Response · Authors · 2024-11-22
> **Responce to Reviewer Km4y (2/2)**
>
> **Q4. Since the proposed HSIC is model-independent, it is suggested that the module's generalization and plug-and-play be verified using more CoOp-based methods.**
>
> DeKg aims to mitigate the bias towards the pre-trained general knowledge caused by the consistency constraint at textual representations in existing knowledge-guided context optimization (KGCO) methods like KgCoOp and TCP. To address this issue, DeKg integrates the independence constraint into KGCO to capture the divergence for intra-class and differentiation for distinct classes between the learnable and crafted prompts. Although PromptSRC obtains good performance, it emphasizes self-consistency on both the image and text sides supplemented by incorporating textual diversity to reduce overfitting in fine-tuning. Therefore, the proposed DeKg just applies to KgCoOp and TCP.
>
> To further investigate the simplicity and effectiveness of DeKg, we apply Dekg to PromptSRC, i.e., DeKg$_\text{PromptSRC}$. As you can see from the comparison shown below, our approach narrowly beats PromptSRC. The main reason is that PromptSRC avoids bias towards pre-training by regularizing through text diversity, which weakens the role of the proposed independence constraint.
>
> Table 1: Comparison of PromptSRC and DeKg$_\text{PromptSRC}$ methods on the base-to-new generalization.
> | Dataset   |      | ImageNet | Caltech101 | OxfordPets | StandfordCar | Flowers | Food101 | FGVCAircraft | SUN397 | DTD   | EuroSAT | UCF101 | Avg.  |
> |-----------|------|----------|------------|------------|--------------|---------|---------|--------------|--------|-------|---------|--------|-------|
> | PromptSRC | Base |   77.60   |    98.10    |    95.33   |     78.27    |  98.07  |  90.67  |     42.73    |  82.67 | 83.37 |   92.90  |  87.10  | 84.26 |
> |           |  New |   70.73  |    94.03   |    97.30    |     74.97    |   76.50  |  91.53  |     37.87    |  78.47 | 62.97 |   73.90  |  78.88 |  76.10 |
> |           |   H  |   74.01  |    96.02   |    96.30    |     76.58    |  85.95  |   91.10  |     40.15    |  80.52 | 71.75 |  82.32  |  82.79 | 79.97 |
> | DeKg$_\text{PromptSRC}$ | Base |   77.60   |    98.17   |    95.13   |     78.03    |  97.63  |  90.75  |     42.58    |  82.59 | 83.53 |  93.38  |  87.02 | 84.22 |
> |    |  New |   70.52  |    93.82   |    96.76   |     75.55    |  77.49  |  91.51  |     37.55    |  78.84 |   63.00  |  75.36  |  78.73 | 76.28 |
> |           |   H  |   73.89  |    95.95   |    95.94   |     76.77    |   86.40  |  91.13  |     39.91    |  80.67 | 71.83 |  83.41  |  82.67 | 80.06 |
>
>
> Thanks for your helpful suggestion.

---

### Official Review · Reviewer_u22y · 2024-11-04

**Soundness:** 3
**Presentation:** 3
**Contribution:** 3
**Rating:** 6
**Confidence:** 4

**Summary:**

This paper introduces a simple yet effective knowledge-based prompt tuning method that leverages the Hilbert-Schmidt Independence Criterion (HSIC) to regularize learnable prompts. By reducing the reliance on prior general knowledge, this approach enables the prompts to better align with task-specific knowledge. The method is versatile and can be easily integrated into other frameworks. When applied to the TCP method, it demonstrates superior performance across most datasets.

**Strengths:**

1. This paper is well-organized and easy to follow. Figure 2 effectively illustrates the main idea by clarifying the roles of each loss function: the \( L_{CE} \) loss enforces alignment between text and vision embeddings, the \( L_{kg} \) loss encourages the learnable prompts to align closely with the CLIP textual embeddings, and the core \( L_{HSIC} \) loss ensures independence within the learnable prompt embeddings.

2. The experiments are comprehensive, covering base-to-new generalization, cross-dataset generalization, and few-shot classification. The proposed DeKgTCP method achieves superior results across most datasets.

**Weaknesses:**

1. Why was the proposed method applied to KgCoOp and TCP rather than other state-of-the-art methods, such as PromptSRC, which performs even better than KgCoOp? Is it more challenging to integrate with PromptSRC, or are the results less effective? Providing additional clarification on this choice would enhance the paper.

2. Figure 4 provides an insight into how the proposed method balances dependence and independence; however, the paper lacks further analysis on this. Expanding on this point would strengthen the reader’s understanding of the method's underlying mechanics.

**Questions:**

see weakness

---

> ### Author Response · Authors · 2024-11-22
> **Response to Reviewer u22y**
>
> Thank you for your encouraging and helpful suggestions. Below, we address the comments you provided.
>
> **Q1. Why was the proposed method applied to KgCoOp and TCP rather than other methods like PromptSRC?**
>
> DeKg aims to mitigate the bias towards the pre-trained general knowledge caused by the consistency constraint at textual representations in existing knowledge-guided context optimization (KGCO) methods like KgCoOp and TCP. To address this issue, DeKg integrates the independence constraint into KGCO to capture the divergence for intra-class and differentiation for distinct classes between the learnable and crafted prompts. Although PromptSRC obtains good performance, it emphasizes self-consistency on both the image and text sides supplemented by incorporating textual diversity to reduce overfitting in fine-tuning. Therefore, the proposed DeKg just applies to KgCoOp and TCP.
>
> To further investigate the simplicity and effectiveness of DeKg, we apply Dekg to PromptSRC, i.e., DeKg$_\text{PromptSRC}$. It can be seen from the comparison shown below, our approach narrowly beats PromptSRC. The main reason is that PromptSRC avoids bias towards pre-training by regularizing through text diversity, which weakens the role of the proposed independence constraint.
>
> Table 1: Comparison of PromptSRC and DeKg$_\text{PromptSRC}$ methods on the base-to-new generalization.
> | Dataset   |      | ImageNet | Caltech101 | OxfordPets | StandfordCar | Flowers | Food101 | FGVCAircraft | SUN397 | DTD   | EuroSAT | UCF101 | Avg.  |
> |-----------|------|----------|------------|------------|--------------|---------|---------|--------------|--------|-------|---------|--------|-------|
> | PromptSRC | Base |   77.60   |    98.10    |    95.33   |     78.27    |  98.07  |  90.67  |     42.73    |  82.67 | 83.37 |   92.90  |  87.10  | 84.26 |
> |           |  New |   70.73  |    94.03   |    97.30    |     74.97    |   76.50  |  91.53  |     37.87    |  78.47 | 62.97 |   73.90  |  78.88 |  76.10 |
> |           |   H  |   74.01  |    96.02   |    96.30    |     76.58    |  85.95  |   91.10  |     40.15    |  80.52 | 71.75 |  82.32  |  82.79 | 79.97 |
> | DeKg$_\text{PromptSRC}$ | Base |   77.60   |    98.17   |    95.13   |     78.03    |  97.63  |  90.75  |     42.58    |  82.59 | 83.53 |  93.38  |  87.02 | 84.22 |
> |    |  New |   70.52  |    93.82   |    96.76   |     75.55    |  77.49  |  91.51  |     37.55    |  78.84 |   63.00  |  75.36  |  78.73 | 76.28 |
> |           |   H  |   73.89  |    95.95   |    95.94   |     76.77    |   86.40  |  91.13  |     39.91    |  80.67 | 71.83 |  83.41  |  82.67 | 80.06 |
>
> **Q2. Figure 4 lacks further analysis, which would strengthen the reader's understanding of the method's underlying mechanics.**
>
> We provide detailed analysis to clarify the insight, i.e., balancing the dependence and independence between learnable and crafted prompts through independence constraint $\mathcal{L}_{kd}$.
>
> As shown in Figure 4, the HSIC values obtained from the consistency-constrained method KgCoOp are very high. This indicates that the learnable tokens are highly correlated with the pre-trained general knowledge, which can lead to poor performance on target tasks. In contrast, the HSIC values derived without knowledge-guided method CoOp are very low. This suggests a weak reliance on general knowledge and a tendency to overfit the target task, resulting in limited generalization ability for target tasks. The values obtained by DeKg are moderate compared to the baselines, indicating a balanced relationship between dependence and independence on general knowledge. This suggests that the HSIC regularization term $\mathcal{L}_{kd}$ introduced in our proposed approach effectively penalizes excessive dependence on learnable prompts while enhancing the adaptability to capture task-specific knowledge.
>
> Thank you again for your valuable feedback. If you have any additional questions or suggestions, we would be happy to address them.

---

### Public Comment · ~Kaixiang_Chen1 · 2025-03-06
**A question**

Thank you very much for your excellent explanation. I truly appreciate your effort and insights.  However, I have a question regarding the formulation of  $\mathcal{L} _ {kd}$. It seems that $\mathcal{L} _ {kd} = \sum_{i} \mathcal{L} _ {kd}(w_i, w_i^{clip})$, where is $\mathcal{L} _ {kd}(w_i, w_j^{clip})$  involved in $\mathcal{L} _ {kd}$?

---

> ### Public Comment · ~Yilun_Li2 · 2025-03-12
> **Response to Kaixiang**
>
> Thank you for your interest in our work and for highlighting the point regarding the formulation of ${L}_{kd}$.
>
> The detailed expansion concerning ${L}_{kd}$ is as follows:
>
> $$
> L_{kd}(\mathbf W, \mathbf W^{clip})={(N_c-1)}^{-2}\sum_i [ \mathbf K  \mathbf H \mathbf K^{clip} \mathbf H]_{ii}
> $$
>
> $$
> =(N_c-1)^{-2}\sum_i \sum_j {[ \mathbf K  \mathbf H]}_ {ij}   {  [\mathbf K^{clip} \mathbf H]}_ {ji}
> $$
>
> $$
> = (N_c-1)^{-2} \sum_i \sum_j \{\mathbf K_ {i,:}\mathbf  H_ {:,j}\} \{\mathbf K^{clip}_ {j,:}\mathbf H_ {:,i}\}                              \qquad                  (1)
> $$
>
> where $\mathbf K_{i,j}=k(\mathbf w_i,\mathbf w_j)$ and $\mathbf K^{clip}_ {i,j}=k(\mathbf w^{clip}_ i,\mathbf w^{clip}_ j)$. Consenquently, $ \mathbf K_ {i,:} \mathbf H_ {:,j} = \sum_ l  \mathbf  K_ {i,l}  \mathbf H_ {l,j} = \sum_l k(\mathbf w_i,\mathbf w_l)  \mathbf H_{lj}$, $ \mathbf K^{clip}_ {j,:} \mathbf H_ {:,i} = \sum_ l  \mathbf  K^{clip}_ {j,l}  \mathbf H_ {l,i} = \sum_l k(\mathbf w^{clip}_ j,\mathbf w^{clip}_ l)  \mathbf H_ {li}$. Thus, it can be deriverd from Eq.(1) as follows
>
> $$
> L_{kd}(\mathbf W, \mathbf W^{clip})= (N_c-1)^{-2} \sum_{i} \sum_j \sum_l k(\mathbf w_i,\mathbf w_l)  \mathbf H_{lj}  k(\mathbf w^{clip}_ j,\mathbf w^{clip}_ l)   \mathbf H_ {li}.
> $$
>
> I hope this explanation resolves the confusion. Should you require any further elaboration or have additional queries, please do not hesitate to reach out.

---

### Meta-Review · Area_Chair_WAC3 · 2024-12-20

**Metareview:**

**Summary:**

This paper proposes a new prompt tuning method by leveraging the Hilbert-Schmidt Independence Criterion (HSIC) as a regularizer. The method is simple and effective. Also, the regularization can be integrated in other frameworks. The authors demonstrated the effectiveness of the proposed method across various datasets. The proposed method learns soft prompts, encouraging the independence between learnable and crafted prompts while maintaining consistency with general knowledge.

**Strengths:**

1. **Simple and effective approach.** The proposed method utilizes an interesting regularization using the Hilbert-Schmidt Independence Criterion (HSIC) to encourage the independence between learnable and crafted prompts.
2. **Versatility/Plug-and-play module.**  The method can be easily integrated into other prompt tuning methods.
3. **No computational overhead.** The proposed method is a learning strategy that does not cause any computational overhead during testing.

**Weaknesses:**

1. **No or marginal improvement with other state-of-the-art methods.**  As Reviewer u22y pointed out, the effectiveness of the proposed method should be evaluated on stronger methods such as PromptSRC. The authors provided additional experimental results in this setting but the improvement is marginal and some degradation was observed on base classes.
2. **Some inconclusive experimental results.** As Reviewer Km4y mentioned, some experimental results are inconclusive. In addition, the proposed methods DeKg and HSIC seem somewhat coupled. A more thorough analysis is needed

**Main reasons:**

The proposed method is simple and effective. Additionally, it is a plug-and-play method, which can be incorporated into other prompting methods. Overall, this paper is well-written and provides sufficient contributions. For these reasons, this paper is recommended for acceptance.

**Additional Comments On Reviewer Discussion:**

The authors provided detailed responses during rebuttal. Although none of reviewers explicitly responded to the authors’ feedback, many concerns raised by reviewers are addressed.

---

### Decision · Program_Chairs · 2025-01-22

Accept (Poster)